



# A Numerical Investigation of Multirotor Systems with Vortex-Generating Modes for Regenerative Wind Energy: Validation Against Experimental Data

Flavio Avila Correia Martins[1], Alexander van Zuijlen[1,‡], and Carlos Simão Ferreira[1]

[1]Faculty of Aerospace Engineering, Flow Physics and Technology Department, Wind Energy Section ([‡]Aerodynamics Section). Delft University of Technology, Kluyverweg 1, Delft, The Netherlands

**Correspondence:** Flavio Avila Correia Martins (f.m.martins@tudelft.nl)

**Abstract.** The current work describes and assesses multirotor systems consisting of pared multirotor and rotor-sized wings, dubbed atmospheric boundary layer (ABL)-control devices, in the rotor's near wake region. The ABL-control devices create vortical flow structures that can accelerate the vertical momentum flux from the flow above the wind farm into the wind farm flow, thus augmenting the wake-recovery process. Understanding the wake-wide impacts of this novel ABL-controlling strategy is crucial to determining the feasibility of using such a strategy in utility-scale wind farms. This work provides numerical assessments of a single multirotor system accompanied by different ABL-controlling setups. The wind flow is modeled via steady-state Reynolds-averaged Navier–Stokes computations. The multirotor and ABL-controlling devices are modeled using three-dimensional actuator surface models based on the Momentum theory. Input force coefficient data for the actuator surface models and validation data for the numerical computations were measured from a scaled model at TU Delft's Open Jet facility. The performance of the ABL-controlling devices was assessed via the net momentum entrained from the flow above the wind farm flow and the total pressure and power available in the wake. It was found that when the ABL-controlling strategy is adopted, the vertical momentum flux becomes the primary mechanism for wake recovery such that for configurations with two or four ABL-controlling wings, the total wind power in the wake recovers 95% of the free-stream value at locations as early as $x/D \approx 6$ downwind of a multirotor system, which is about one order of magnitude faster than what is seen for the baseline wake without ABL-controlling capabilities.

## 1 Introduction

Our energy transition goals require a significant increase in installed wind power capacity, typically achieved by expanding the number and size of wind farms. However, scaling up wind farms, both onshore and offshore, presents various challenges, including technical, environmental, economic, and social acceptance issues. For example, large onshore wind farms can create conflicts with nearby residents due to noise and visual pollution (Zerrahn, 2017; McKenna et al., 2015). Likewise, large offshore wind farms often face high operational and energy transmission costs (Sadorsky, 2021). To address the need for larger wind farm areas, we can improve the ratio of total power output per land surface area by enhancing the wake-recovery process. In





this study, we evaluate multiple configurations of a novel atmospheric boundary layer (ABL) control strategy (Ferreira et al., 2024) that boosts vertical momentum flux in a wind farm, thereby increasing the total power output per land surface area.

The need for larger wind farms is closely linked to the spacing required for effective wake recovery between consecutive wind turbines (see top diagram in Fig. 1). With the characteristic height of the ABL around one kilometer, wind farms covering a surface area of over 1020 km can approach the asymptotic limit of "infinite" wind farms. In this scenario, the boundary layer flow may reach a fully developed state, where most kinetic energy entrainment occurs above the farm (Abkar and Porté-Agel, 2013). In such cases, velocity fluctuations become the primary source of kinetic energy. Since momentum transfer via velocity

fluctuations is typically slow, there is a clear need for better strategies to accelerate wake recovery. The most common strategies can generally be categorized into three types: i. wing-pitch control, ii. yaw control, and iii. torque control strategies.

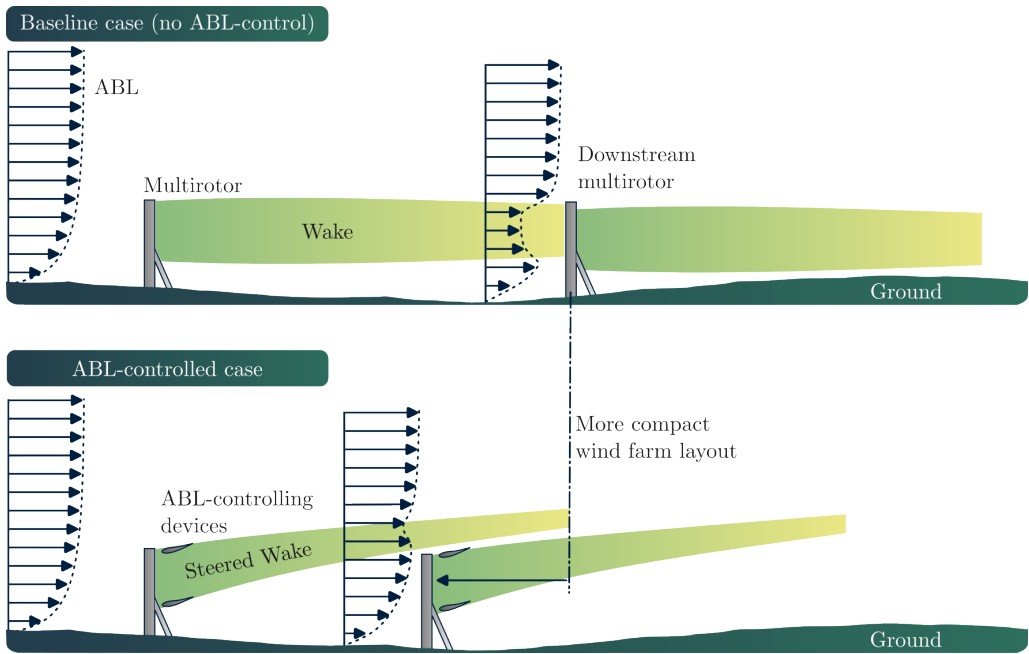

**Figure 1.** The figure shows a schematic illustration of the wake behind a multirotor system before (top) and after (bottom) implementing ABL-control systems. By directing the wake upward into the atmosphere, ABL-controlled wind farms can achieve higher power output per unit of land area, allowing them to be more compact.

The central concept of wing-pitch control strategies is to steer the wake away from downwind turbines (Dilip and Porté-Agel, 2017). Wake recovery via blade-pitch control is a well-explored topic in the literature, commonly based on blade-element-momentum theory for rotor aerodynamics (Nash et al., 2021). Ferreira (2009) examined how typical H- and W-type

vertical-axis wind turbines (VAWT) generate two counter-rotating tip vortices at the ends of the blades, which sustain more accelerated wake deflection in comparison with non-pitched wings (Tescione et al., 2014; Ryan et al., 2016; Wei et al., 2021). Ferreira proposed that by shifting the VAWT's pitching axis (along the blade's quarter-chord length), additional cross-wind momentum could be added to the incoming streamlines, thereby speeding up wake deflection. His work has inspired similar





numerical and experimental research on VAWT wake deflection via pitched blades. Jadeja (2018) investigated the topologies of wakes deflected by pitched VAWTs using actuator line models with unsteady Reynolds-averaged Navier-Stokes (RANS) simulations. Jadeja noted that the wake-deflection strategy via VAWT blade pitching has the advantage of not affecting the upstream turbine's performance, unlike yaw control strategies used for horizontal-axis wind turbines. Huang (2023) measured wake deflections of an H-shaped VAWT at different pitching angles and observed that blade-tip vortices could effectively double the rate of lateral wake deflection through active pitch control.

Many studies also focus on improving wind farm power output through yaw-control strategies (Howland et al., 2019). In VAWT farms, wake deflection is usually achieved by altering the rotor blade layout to generate momentum-carrying vorticity in the wake (Huang et al., 2023). Yaw and blade-pitch control strategies can also be combined. For instance, Wang et al. (2020) found that adjusting blade pitch can mitigate the excess loads induced by yaw control. Lastly, torque control strategies aim to manage the strength of the turbine wake to increase overall wind farm power production. This approach reduces the induction factor of upstream turbines in exchange to increase the kinetic energy available for downstream turbines (Bartl and Sætran, 2016).

However, the wake-recovery strategies mentioned above have common shortcomings. They require complex changes to wind farm control algorithms, which can conflict with reliability and safety-oriented controls. These strategies can also impose additional loading patterns on conventional turbine designs, potentially leading to premature failures (Wang et al., 2020). Most importantly, they usually penalize the performance of individual turbines to enhance wake reenergizing, hoping the downstream turbine's power gains outweigh the upstream's losses. Consequently, total power production can rarely increase by more than about 30% (Bader et al., 2018) in the best laboratory conditions. For example, Gebraad et al. (2016, 2017) reported a 5% increase in annual energy production at the Princess Amalia Wind Park in the Netherlands using a combined layout optimization and wake steering control strategy. Thus, these methodologies may not offer as much potential for reducing wind farm size in real-life scenarios.

This study proposes a new wake-reenergizing strategy that can significantly extend the energy potential of wind farms. We describe and assess multirotor setups with paired multirotor and rotor-sized wings, referred to as ABL-control devices, placed in the near wake region (see bottom panel in Fig. 1). In this setup, the rotor-sized wings create vortical structures that accelerate vertical momentum flux from above the ABL into the wind farm, increasing net power production. Note that ABL-controlling systems are intended to consist of multiple multirotor setups combined with ABL-controlling mechanisms. However, this paper focuses on evaluating the performance of a single multirotor system equipped with ABL-controlling devices. Multirotor and ABL-control devices are modeled using three-dimensional actuator surface models based on Momentum theory (see Fig. 2). Steady-state RANS computations performed in `OpenFOAM` are proposed to validate this proof-of-concept. The effects of the ABL-control wing's induced drag are also investigated by comparing models with and without the wings' induced drag forces. The performance of the ABL-control devices is quantified by the wind farm's total pressure and vertical momentum flux through the ABL. Experimental observations from a scaled multirotor system set up in TU Delft's Open Jet facility complement the present numerical results.



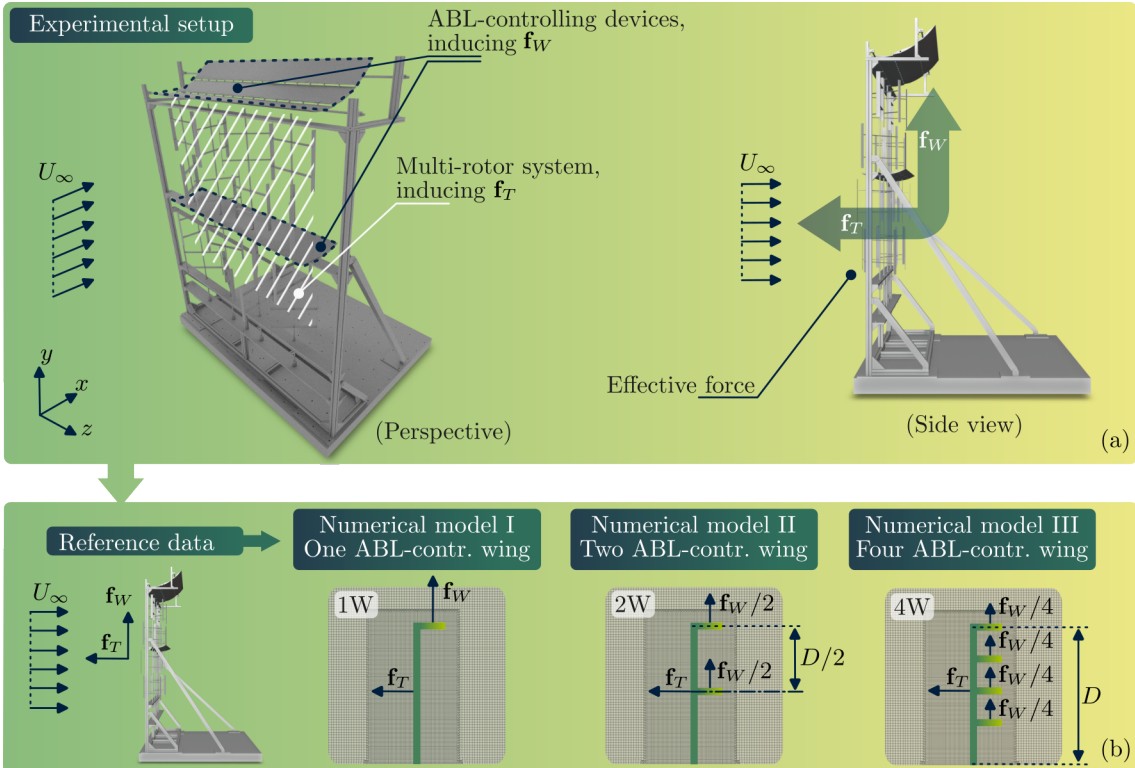

**Figure 2.** Panel (a) shows details of the reference experimental setup (a) in comparison with the numerical model based on momentum theory in Panel (b). The experimentally measured multirotor system's thrust force vector, $\mathbf{f}_T$, and ABL-system's effective aerodynamic force, $\mathbf{f}_W$, are homogeneously distributed in a cell-weighted fashion in the numerical model.

This paper is organized as follows. In § 2, it is presented the governing equations, numerical setup, and the description of the assessed test cases. In § 3 results are discussed. § 3.2, delves into the main flow features of the ABL-controlled flows via analyses of the velocity and vorticity fields for different possible ABL device configurations. § 3.3 concerns assessing the performance of the different ABL-devices layouts via analyses of the momentum fluxes and quantifying the total pressure and power available in the wake.

## 2 Methodology

### 2.1 The numerical model

We model the flow around the actuator surfaces using the steady-state RANS equations for incompressible turbulent flows (Darwish and Moukalled, 2016):



$$\nabla \cdot \overline{\mathbf{u}} = 0 \tag{1}$$

$$(\overline{\mathbf{u}} \cdot \nabla)\,\overline{\mathbf{u}} = -\frac{1}{\rho}\nabla \overline{p} + \nabla \cdot (\nu \nabla \overline{\mathbf{u}}) - \nabla \cdot \mathbf{R} + \frac{1}{\rho}\mathbf{f}_T + \frac{1}{\rho}\mathbf{f}_W, \tag{2}$$

In Eqs. 1 and 2, $\mathbf{u} = u_x\hat{\imath} + u_y\hat{\imath} + u_z\hat{k}$ is the velocity, $p$ is the pressure, $\rho$ is the density, $\nu$, is the kinematic viscosity, $\mathbf{R} = \overline{\mathbf{u}'\mathbf{u}'}$ is the Reynolds-stress-tensor, which couples the mean flow with the turbulence. The overbar, $\overline{\langle \cdot \rangle}$, and the prime, $\langle \cdot \rangle'$, represent the mean and the fluctuating components of the respective scalar or vector variable. In this study, the homogeneously-distributed force $\mathbf{f}_T$ models the effective thrust of the multirotor system, whereas $\mathbf{f}_W$ models the aerodynamic loads associated with the ABL-controlling wings. Notice that due to constant, homogeneously distributed aerodynamic loadings, the current models are relatively steady, and thus, an unsteady model is deemed deferable. Moreover, since the main mechanism for energy transportation relies on the generation and advection of large-scale wing-tip vortical structures, higher-fidelity models, e.g., Large-Eddy simulations, are also deemed futile for this proof-of-concept investigation, where the authors opted for assessing the largest possible number of test cases. As shown in§ 3.1, the current steady RANS model is sufficient for the carried analysis.

The turbulent wind farm flow is modeled using the shear-stress-transport (SST) $k-\omega$ model (Menter et al., 2003) based on an uncertainty assessment conducted by Hornshøj-Møller et al. (2021). The SST model belongs to the linear-eddy-viscosity class of RANS models, which assumes a linear relation between Reynolds stresses and the mean strain rate (i.e., the Boussinesq's hypothesis):

$$\mathbf{R} \approx -2\nu_t \overline{\mathbf{S}} + \frac{2}{3}\mathbf{I}k, \tag{3}$$

where $\nu_t$ is the eddy viscosity, $\mathbf{S} = (\nabla \overline{\mathbf{u}} + (\nabla \overline{\mathbf{u}})^T)/2$, is the mean strain-rate tensor, $\mathbf{I}$ is the second-order identity tensor, and $k := \mathrm{tr}(\mathbf{R})/2$ is the turbulent kinetic energy. The eddy viscosity is computed from the turbulent kinetic energy, $k$, and the specific dissipation rate, $\omega$, which can be estimated using:

$$k = \frac{3}{2}(T_{u,\infty}U_\infty)^2 \tag{4}$$

and

$$\omega = \frac{k^{1/2}}{C_\mu^{1/4}D}. \tag{5}$$





In the expressions above, $T_{u,\infty}$ and $U_\infty$ are the turbulence intensity and reference velocity in the unperturbed, far-field flow, such that $U_\infty := |\overline{\mathbf{u}}_\infty|$; $C_\mu$ is a constant equal to 0.09 and $D$ is the reference length scale. Here, the side length of the multirotor system is adopted as the reference length scale (see Fig. 2).

The effects of the multirotor system and ABL-controlling wings are modeled as body force (Wu and Porté-Agel, 2011), such that the effective forces are uniformly distributed across the finite-volume cells comprising the multirotor and ABL-controlling devices regions, respectively. The distributed multirotor force, $\mathbf{f}_T$, is represented as a source term in the momentum equation in Eq. 2, and it is estimated using:

$$\mathbf{f}_T = \left(\frac{1}{2}\rho D^2 U_\infty^2\right) C_T \hat{\imath}, \tag{6}$$

where $C_T$ is the effective, experimentally-measured thrust-force coefficient. The ABL-controlling wings are likewise modeled as source terms in the momentum equation using:

$$\mathbf{f}_W = \left(\frac{1}{2}\rho A U_\infty^2\right) \left(C_{x,W}\hat{\imath} + C_{y,W}\hat{\imath}\right), \tag{7}$$

where $C_W$ is the wing's effective force coefficient. The force coefficients $C_T$ and $C_W$ are determined from experimental observations of a scaled model at TU Delft's Open Jet Facility. Further details on the experimental facility can be found on (Lignarolo et al., 2014).

## 2.2  Numerical setup

In this work, all numerical computations are performed using `OpenFOAM` v9 (Greenshields, 2023; Weller et al., 1998) jointly with the momentum sources, i.e., $\mathbf{f}_W$ and $\mathbf{f}_T$, computed using a Momentum-theory-based code written by the authors (Martins, 2024). The momentum sources are distributed in a cell-volume weighted fashion onto the cell centers comprising the multibody systems and ABL-controlling wings. The multirotor extends one finite-volume cell in the streamwise direction (i.e., one-cell thick) and is $1D$-long in the spanwise, $z$-direction. The height of the multirotor system region is $1D$, with its base far $y/D = 0.10$ from the ground plane. The ABL-controlling wings are also one finite-volume cell wide in the streamwise direction and are one finite-volume cell thick in the vertical, $y$-directions. The computational domain extends $50D \times 20D \times 10D$ in the downwind, $x$, spanwise, $z$, and vertical, $y$, directions, respectively. The multirotor array is located $10D$ downwind of the inlet. These domain dimensions satisfy all minimum domain-sizing requirements to minimize the boundary effects on the performance of the turbine (Rezaeiha et al., 2017; Gargallo-Peiró et al., 2018). Details of the numerical setup are shown in Fig. 3.

A steady state, incompressible solver (i.e., `simpleFoam`) is selected for the simulations. The SIMPLE method is employed for pressure-velocity coupling. The Gaussian integration was used with different interpolation schemes for the spatial dis-





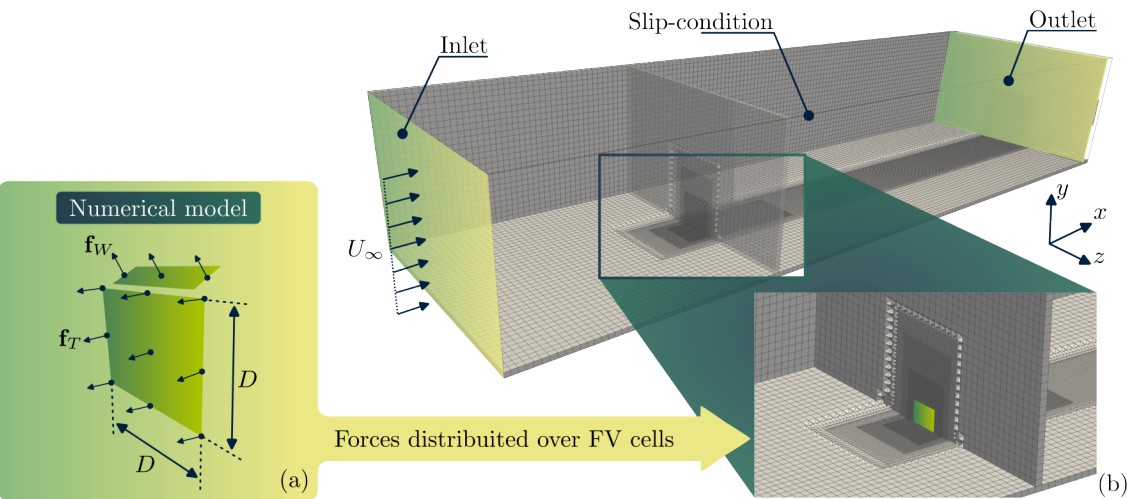

**Figure 3.** Panel (a) shows details of the numerical model based on Momentum theory; panel (b) shows details of the computational domain with the actuator forces modeled as momentum sources. The computational domain is highlighted at the top.

cretization of differential operators. The second-order linear interpolation was employed for gradient terms, the second-order bounded upwind interpolation for divergence terms, and the second-order linear corrected interpolation was employed for the Laplacian terms. The Geometric agglomerated algebraic multigrid (i.e., a V-cycle type) solver was adopted with the Gauss-

Seidel preconditioner method for pressure and its symmetrical version for velocity and turbulence variables. An error tolerance of $1 \times 10^{-6}$ was adopted for all smooth solvers.

Modeling a neutral ABL involves inlet boundary conditions providing log-law type ground-normal inflow boundary conditions for wind velocity and turbulence quantities (Parekh and Verstappen, 2023). However, for this proof-of-concept work, it is deemed more convenient to model the inlet boundary condition with a uniform (Dirichlet) profile for velocity and turbulence-

model quantities and a zero-gradient (Neumann) boundary condition for pressure. This simplification assumption allows for generalizing the current results by eliminating ambiguousness associated with ground-normal velocity profiles. The sides of the domain are modeled as a slip (Neumann) boundary condition. The outlet is modeled as a free-stream flow condition and a zero-pressure boundary condition for pressure. The domain's inlet, top, and side boundaries are modeled with (Dirichlet) inlet boundary conditions for velocity, pressure, and turbulence-model quantities. The bottom surface of the domain is mod-

eled as a slip (Neumann) boundary condition for the velocity and turbulence variables and zero-gradient for pressure. The outlet is modeled as a free-stream flow condition and zero-gradient for pressure. The free-stream wind velocity, $U_\infty$, renders a diameter-based Reynolds number of $Re_D := \rho D U_\infty \mu^{-1}$, of $Re_D \sim 4 \times 10^8$. Turbulence quantities are computed from a baseline free-stream turbulence intensity level of $T_{u,\infty} = 1\%$, verging upon values of the reference dataset collected in TU Delft Open Jet facility.





## 2.3 Cases descriptions

Seven setups are investigated in this study. The setups consist of multirotor systems paired with four, two, one, or no ABL-controlling wings. To isolate the effects of the ABL-system wing's induced drag on the systems' performance, ABL-controlled numerical models are subdivided into twofold: models with or without the wing's induced drag. The thrust coefficient for the numerical model was based on an experimental observation for a scaled multirotor system at a tip-speed ratio of 3.1, resulting in an effective thrust coefficient of approximately $C_T = 0.72$. The effective lift force coefficient of the ABL-controlling system amounts to $C_{y,W} = 0.82$. The effective induced drag of the ABL-controlling system is $C_{x,W} = 0.15$. In all test cases, $\mathbf{f}_T$ and $\mathbf{f}_W$ are kept constant and are homogeneously distributed among all the wings in a cell-weighted fashion, such that $C_{y,W}$ and $C_{x,W}$ are the same throughout all cases. The summary of the cases contemplated in the current study and the corresponding nomenclature is highlighted in Table 1 (note that the authors provided complementary test cases in (Martins et al., 2024)).

**Table 1.** Summary of the relevant operational parameters of the analysed systems.

| Case name | ABL-devices force coefficients $(C_{x,W}\hat{\imath}, C_{y,W}\hat{\imath})$ | Number of ABL-controlling wings |
|---|---|---|
| Baseline | (0, 0) | 0 |
| 1W | (0.15, 0.72) | 1 |
| 2W | (0.15, 0.72) | 2 |
| 4W | (0.15, 0.72) | 4 |
| 1W-ND | (0, 0.72) | 1 |
| 2W-ND | (0, 0.72) | 2 |
| 4W-ND | (0, 0.72) | 4 |

## 3 Results

### 3.1 Grid-independence analysis and model validation

The sizing of the cell elements was set to minimize the grid-sizing influence on the total pressure, $p + \overline{\mathbf{u}}^2 \rho / 2$. Results are considered grid-independent if the total pressure sampled at different planes downwind of the turbines is insensitive (less than 1% different) to further grid refinements. Grid refinements were performed by halving the diameter of the cell elements, $\Delta l$, between two consecutive grid refinements. From the grid-independence analysis, it was found that a grid with cell sizings of $\Delta l/D \approx 0.03$ in the near-wake region ($4D \times 2D \times 2D$ wide box centered on the turbine); $\Delta l/D \approx 0.06$ along the wake and, $\Delta l/D \approx 0.27$ in the far-field was sufficient for grid-independent results. The resulting computational grid, consisting of $\approx 2.4 \times 10^6$ cells, is schematically illustrated in Fig. 3.

Subsequent to the grid-independence analysis, the present numerical model is validated by comparing the model results to experimental data collected at TU Delft's Open Jet facility at $Re_D \sim 3.8 \times 10^5$. In Figs. 4 and 5, the numerical and measured $u_x$-colored wake profiles at $x/D = 1$ (i.e., one diameter downstream of the multirotor system) are shown. The current numerical and experimental setups consist of a multirotor system paired with a double ABL-controlling system. One ABL-





**Table 2.** Results of the grid-independence analysis. The "Fine" mesh is adopted in this study. The relative error refers to the normalized error between two consecutive grid refinements.

| Mesh | Size of smallest grid element $\Delta l/D$ | Number of finite-volume cells | Relative error in total wake pressure, $p_t$ |
|---|---|---|---|
| Coarse | 0.07 | $1.9 \times 10^5$ | - |
| Medium | 0.05 | $5.4 \times 10^5$ | 0.06% |
| Fine | 0.03 | $2.4 \times 10^6$ | 0.03% |
| Dense | 0.02 | $7.6 \times 10^6$ | 0.03% |

controlling wing is at $y/D = 1.25$, whereas the second is at $y/D = 0.75$. The reference velocity consists of time-averaged data obtained through Particle Image Velocimetry. Further details of the experimental setup can be read in (Bensason et al., 2024)

and (Broertjes et al., 2024).

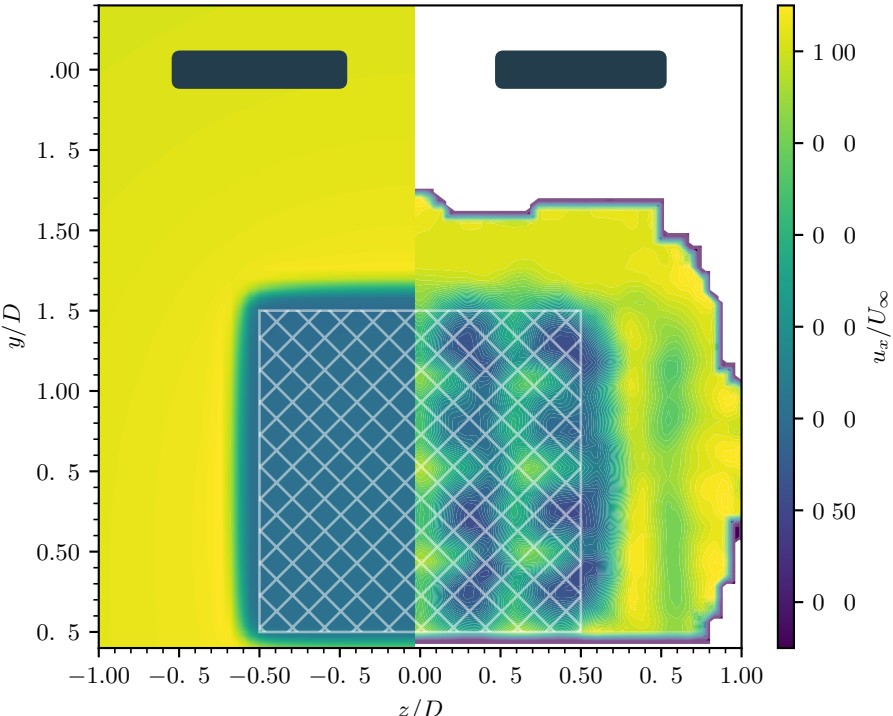

**Figure 4.** Comparison of simulated flow (left-hand side) and experimental flow (right-hand side) behind a disk actuator model and the multirotor setup, respectively (rotor projected area represented by the white-hashed region). The figure displays $u_x$-velocity colored fields at $x/D = 1$. No ABL control was applied. The experimental velocity field was reconstructed from time-averaged PIV data (refer to (Bensason et al., 2024) and (Broertjes et al., 2024)), while the numerical results were obtained in this study through RANS computations.

Figure 4 shows the comparison of the $u_x$-colored velocity fields at $x/D = 1$ without the ABL-controlling devices. Notice that the numerical velocity field yields a uniform induction field, $a$, $a := (U_\infty - u_x(x = 0))/U_\infty$, behind the multirotor's region (rotor projected area represented by the white-hashed region) due to the homogeneously distributed thrust coefficient based on





Momentum theory. The experimental results show, in contrast, velocity fields with local fluctuations due to the discrete nature
of the scaled multirotor array. The experimental setup comprises a multirotor system with a $4 \times 4$ vertical-axis rotors array.
Nonetheless, the results shown in the figure uphold the strong correlation between the current numerical and the reference
experimental data.

In Fig. 5, the validation exercise is extended to account for the effects of the ABL-controlling wings. The test case in Fig. 5
consists of a setup with an actuator wing at the top of the multirotor device ($y/D = 1.1$) and a second actuator wing at the
185 center of the device (i.e., $y/D = 0.75$). The figure also shows the $u_x$ fields at $x/D = 1$. Once again, there is a strong correlation
between the experimental and numerical results, evident from the induction field behind the multirotor system and the size and
shape of the two wing-tip vortices at $y/D \approx 1.35$.

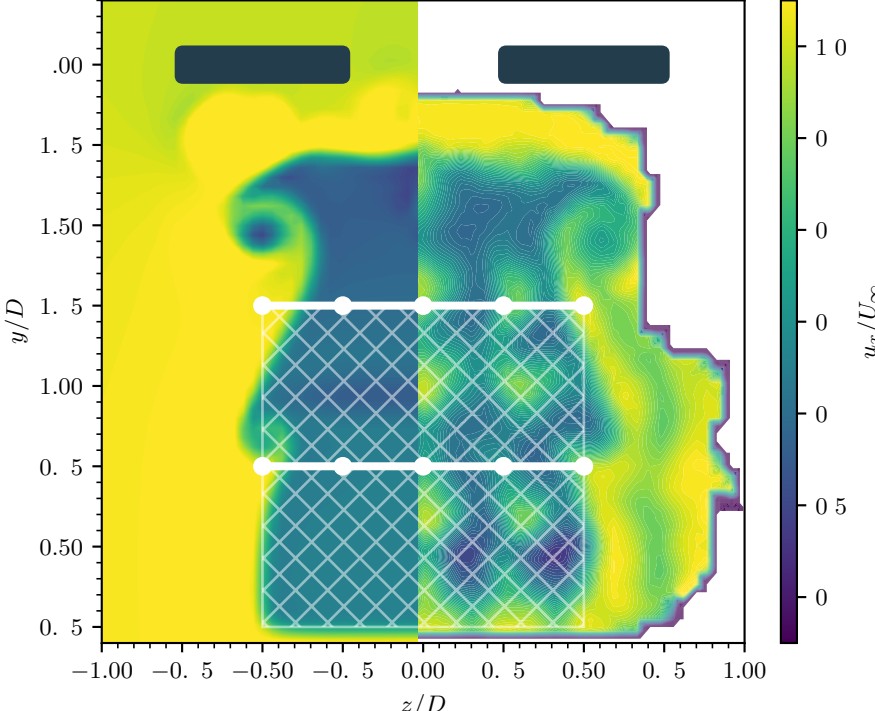

**Figure 5.** Comparison between simulated flows (on the left-hand side) and experimental flows (on the right-hand side) behind the ABL-
controlled setups (rotor projected area represented by the white-hashed region). The ABL-controlling devices are represented by the dotted
line segments. The figure displays $u_x$-velocity colored fields at $x/D = 1$. The experimental velocity field was reconstructed from time-
averaged PIV data (refer to (Bensason et al., 2024)), while the numerical results were obtained in this study through RANS computations.
Visual inspection of the figure highlights a strong correlation between the numerical and reference experimental results.





## 3.2 ABL-controlled wake characteristics

Figure 6 presents the wake behind the multirotor system as streamwise-velocity colored fields, $u_x/U_\infty$, at various planes
downstream of the multirotor system (with the origin of the coordinate system, i.e., $x/D = 0$, at the multirotor's location). In all
panels, the silhouette of the multirotor system is overlaid in white for reference. The ABL-controlling devices are represented
by the dotted line segments. The top row illustrates the wake of the Baseline case without ABL-controlling features. The
subsequent rows depict results for increasing numbers of ABL-controlling devices, labeled as 1W-ND' (one wing, no induced
drag) for one wing, 2W-ND' for two wings, and '4W-ND' for four wings. The effects of the wing's induced drag are not
considered in these cases.

In cases with ABL-controlling devices, an overall ascent motion with crossflow expansion characterizes the general dy-
namics of the ABL-controlled wakes. Figure 6 also demonstrates that, as expected, the rapid ascent of the wakes is facili-
tated by the counter-rotating, wing-tip vortices of Cases 1W-ND, 2W-ND, and 4W-ND. Additionally, for Case 1W-ND, fluid
parcels near the middle plane $z/D = 0$ are observed to ascend more rapidly compared to those near the two wing-tip vortices
(around the $z/D \approx \pm 0.5$ planes), which ascend to the flow above the turbine ($y/D > 1.10$) at lower speeds. In the absence
of ABL-controlling devices (i.e., the Baseline case), wake re-energizing relies solely on momentum exchange via velocity
fluctuations on the outer shear layers of the wake, which is notably less efficient than the wake-steering technique evaluated in
this study. This observation is evident from a visual examination of the volume of the high-induction flow region that remains
at $y/D < 1.10$ for the different cases. Henceforth, we refer to the flow region below the $y/D = 1.10$ plane as the 'wind-farm
flow'. The authors also note that these conclusions are applicable to flows with higher free-stream turbulence intensity levels,
provided that large vortical structures are not dissipated or depleted due to flow instabilities.

The velocity fields in Fig. 6 indicate that the advection of the high-induction fluid parcels behind the multirotor system to
the flow above the wind-farm flow is considerably improved when designs with two or more wings are adopted in comparison
with an arrangement with a single wing. The extra efficiency of Cases 2W-ND and 4W-ND compared to Case 1W-ND can be
noted from the visual inspection of the wake's topology. In Cases 2W-ND and 4W-ND, the velocity-deficit region defining the
wake that remains in the wind-farm flow for $x/D > 5$ is substantially narrower compared to what is seen for Case 1W-ND.
Nonetheless, despite the overall more efficient upwards advection of the wake promoted by the extra wings of Cases 2W-ND
and 4W-ND, the velocity-deficit flow is, on average, lower in the ABL in these cases when compared to the results for Case
1-ND. For instance, at the downstream plane $x/D = 7$, the maximum height of the wake is $y/D \approx 2.6$ for Case 1W-ND,
whereas for Cases 2- and 4W-ND, the maximum height of the wake is $y/D \approx 2.2$. The authors remind the reader that the total
vertical force is the same in all ABL-controlled cases.

Figure 7 extends the results shown in Fig. 6 to account for the Blade's induced drag (see Table 1). The comparison of the
setups without (i.e., Cases 1W-ND, 2W-ND and 4W-ND, shown in Fig. 6) against the results with (i.e., Cases 1W, 2W, and 4W,
shown in Fig. 7) induced drag reveal that the main flow features of the ABL-controlled systems, such as the vortexes formation,
shedding and advection, are not significantly affected by the induced drag. This last conclusion is especially true for the cases
with more wings, where the effective drag force is distributed over a larger flow region. For the Case 1W, for which results

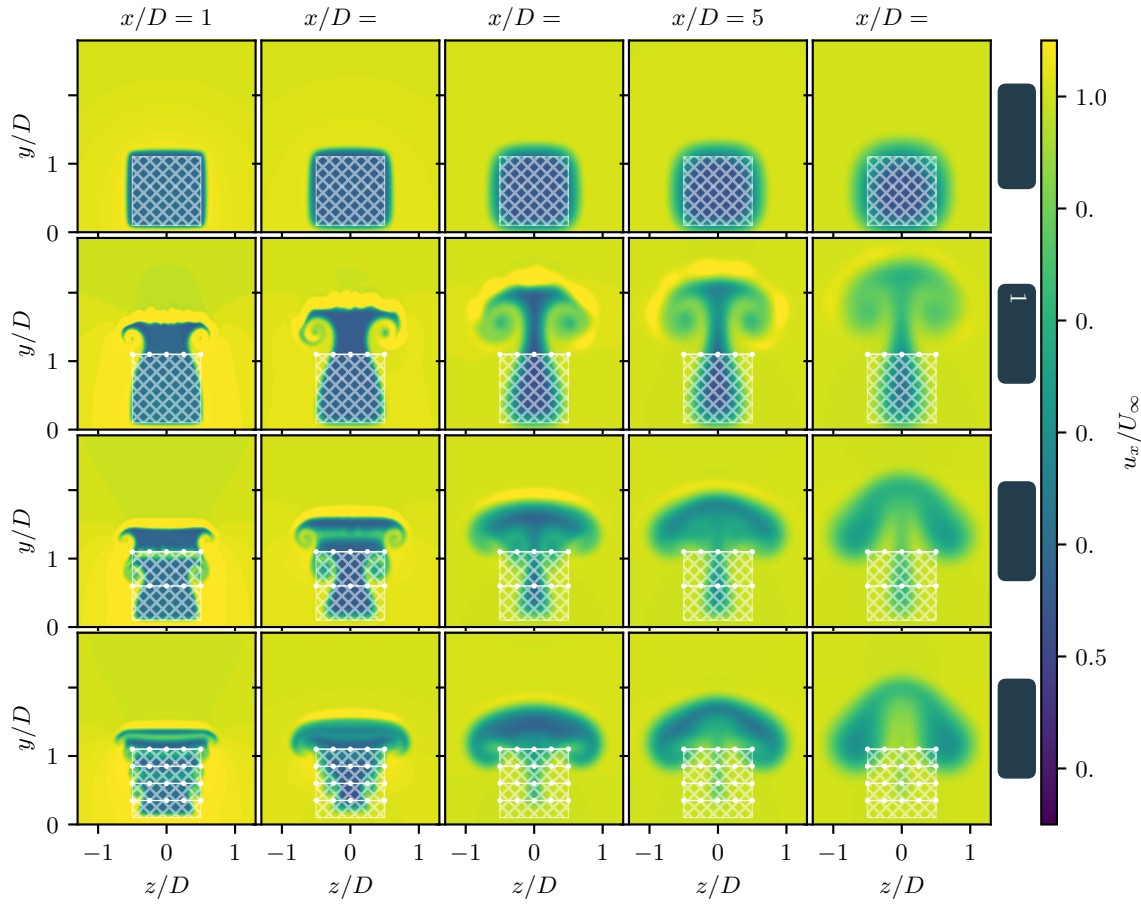

**Figure 6.** The figure shows the streamwise-velocity contours at different downwind locations, $x/D$, behind the multirotor system (rotor projected area represented by the white-hashed region). The top row shows the conventional squared wake of the baseline case without the ABL-controlling devices. Subsequently, the second, third, and fourth rows show results for the ABL-controlled system with 0, 1, 2, and 4 wings (represented by white-dotted line segments), respectively. These simulations do not account for the effects of the wing's induced drag.

are shown on the second row of Fig. 7, the concentrated drag force induced on the flow by the single ABL-controlling wing breaks up instabilities over the outer shear layers of the wake. Such instabilities, which originated due to the local curvatures in the induction field behind the drag-inducing wing, are more pronounced on the near-wake region, i.e., at $x/D < \sim 4$, and are quickly diffused under the action of viscosity.

Wind farms with ABL-controlling devices feature enhanced momentum exchange between the wind farm flow and the flow above through wing-tip vortices. Thus, to fully understand the flow mechanisms promoting the upwards motion of the wake, and more specifically, to understand why the momentum-deficit fluid parcels are advected upwards at different rates depending on the number of wings and their location across the distinct $z/D$ planes, the visual diagnostics of the velocity fields are insufficient. Instead, assessing the flow through the vorticity fields is essential. Figure 8 shows, top-to-bottom, the downwind-vorticity colored, $\omega_x D/U_\infty$, cross-sectional planes of the flow for increasing number of ABL-controlling wings.

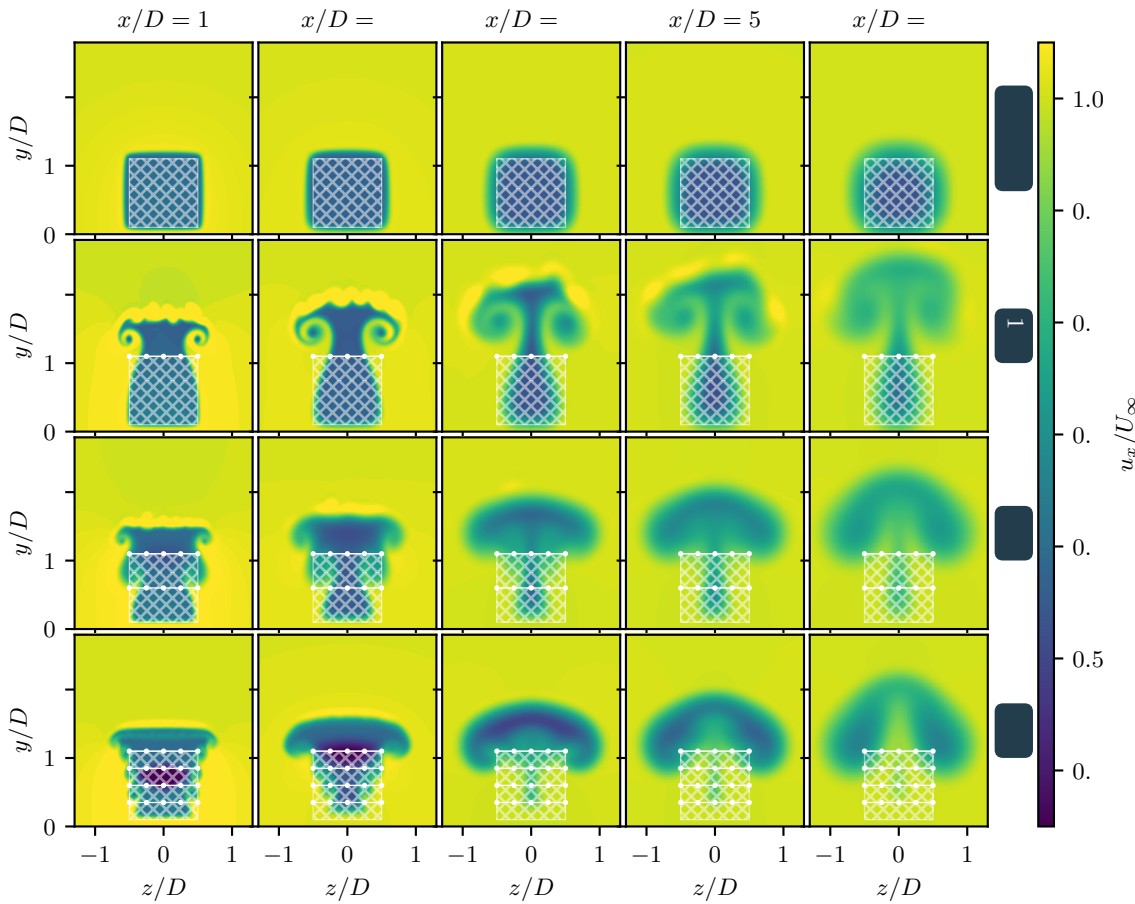

**Figure 7.** This figure extends the results shown in Fig. 6, accounting for the effects of the wing's induced drag. Each panel shows the streamwise-colored wake behind the multirotor system (rotor projected area represented by the white-hashed region) at different downwind locations, $x/D$. Each row shows, from top-to-bottom, results for the ABL-controlled system with 1, 2, and 4 wings, respectively (represented by white-dotted line segments).

Case 1W-ND's vorticity-colored fields, shown in the top row of Fig. 8, reveal the two counter-rotating wing-tip vortices developed behind the multirotor system by the single ABL-controlling wing. These two wing-tip vortices are advected upwards and cause the upward motion of wake the closer the flow parcels are to the symmetry plane of the flow, $z/D = 0$, while
simultaneously inducing the downwash motion of the high-momentum flow above the wind-farm flow at $|z/D| > 0.5$. For Cases 2W-ND and 4W-ND, an average upward movement of the wake is also observed. However, the mechanisms of the upward flow advection are considerably different in Cases 2W-ND and 4W-ND compared to Case 1W-ND and require an orderly analysis of the events.

To better understand the ABL-controlled flow of Cases 2W-ND and 4W-ND, let us first subdivide the advection of the wake in
Cases 2W-ND and 4W-ND into threefold based on the steps of the upwards-motion mechanisms: i. $x/D \in \sim [0,2]$, ii. $x/D \in \sim [2,5]$ and iii. $x/D \in \sim [5,\infty[$. The authors will denote here, for simplicity, step i. as "vortex-formation"; step ii. as "vortex-







**Figure 8.** Streamwise-vorticity contours reveal the wing-tip vortexes generated by the different ABL-controlling device designs. Top-to-bottom rows show results for 1, 2, and 4 ABL-controlling wings (represented by black-dotted line segments), respectively.

coalescing" and iii. as "vortex-advection" step. During the vortex-formation step, the wing-tip vortex is advected downstream by the mean flow. Concomitantly, the vortices that are above the barycenter of the vortical system (i.e., the horizontal line





equally apportioning the vortices above it with the vortices below it) are pushed outwards from the symmetry plane $z/D = 0$
(see Fig. 8), whereas the vortices below the barycenter of the vortical system are pushed inwards, i.e., towards the plane
$z/D = 0$. Notice also that these lateral motions of the vortical system are self-induced. During the vortex-coalescing process,
the low-pressure regions correspondent to the core of the wing-tip vortices act as attracting regions, and the multiple wing-tip
vortices coalesce into two skewed, counter-rotating vortical structures (see corresponding panels for $x/D = 4$ in Fig. 8). In the
third and last step, i.e., for $x/D > 5$, the induction fields of one vortex on another mutually propel the two vortical systems
upwards into the atmosphere. Notice that for Case 1W-ND, this third step co-occurs with the vortex formation process, thus
explaining why the upwards-advection of the vortex paired in Case 1W-ND takes significantly less time to move to the flow
above the wind farm flow, as observed from Fig. 7.

Figure 8 includes vorticity-fields of setups accounting for the ABL-controlling devices's induced drag, i.e., Cases 1W, 2W,
and 4W. The comparison of the streamwise vorticity fields for the cases with and without the effects of induced drag reveals
that, despite the considerable magnitude of $C_{x,W}$, the dynamics of the wing-tip vortices are relatively unaltered in the presence
of the induced drag effects. Nonetheless, it is evident that when drag forces are accounted for, a surplus of flow instabilities
is introduced over the topmost region of the wake. Such three-dimensionalities are naturally associated with turbulent mixing
that can ultimately contribute to the mixing between the wind farm flow and the flow above the wind farm.

Given that the wing-tip vortices are the primary mechanism for moving the low-momentum flow of the wake upwards in
exchange for moving the high-momentum ABL flow downwards, a simplified analysis of the flow in the view of the Kutta-
Joukowski theorem is deemed relevant. Let us denote the circulation around a closed contour in an arbitrary crossflow plane, $yz$,
by $\Gamma_x$. If the conservative-flow assumption is adopted, the Kutta-Joukowski theorem expresses that the circulation associated
with the wing-tip vortices of Case 1W (or 1W-ND) can be approximated by:

$$\Gamma_x = -C_y U_\infty \frac{D}{2}. \tag{8}$$

Moreover, it is also known from potential flow theory that the tangential velocity $u_\theta$ induced by a point-vortex idealization at
a flow parcel distant $r$ from the same point-vortex is $u_\theta = -\Gamma_x/2\pi r$. Thus, if we approximate the current flows to a conservative
flow idealization, it is straightforward that the vertical velocity $u_y \approx u_y(z/D)$ induced at the symmetry plane $z/D = 0$ of the
flow, $u_y(z/D = 0)$, due to the two wing-tip vortices for a single ABL-controlling wing is

$$u_y(z/D = 0) \sim 2u_\theta \sim \frac{C_y U_\infty}{\pi} \hat{\imath}. \tag{9}$$

The result above shows that due to the action of the wing-tip vortices, fast upward advection of fluid parcels that scale with
$U_\infty$ is expected near the $z/D = 0$ plane. Contrastingly, the vertical velocity induced on one wing-tip vortex by another scaled
with the distance $D$ is, thus, similar to $u_y(z/D = \pm 1/2) \sim C_y U_\infty/4\pi \hat{\imath}$, i.e.:





$$\frac{u_y(z/D = 0)}{u_y(z/D = \pm 1/2)} \sim 4. \tag{10}$$

The results in Eq. 10 explain why the flow parcels $z/D = 0$ are advected upstream at considerably quicker rates than the
flow parcels in close proximity to the vortical structures themselves. Figure 9, showing the $u_y$-velocity colored flows past
ABL-controlled wakes at $x/D = 7$, confirms the analytical results deduced above. The panels in the figure show, left to right,
results for Cases 1W, 2W, and 4W, respectively. Results for Cases 1W-ND, 2W-ND, and 4W-ND are similar and are omitted
for conciseness.

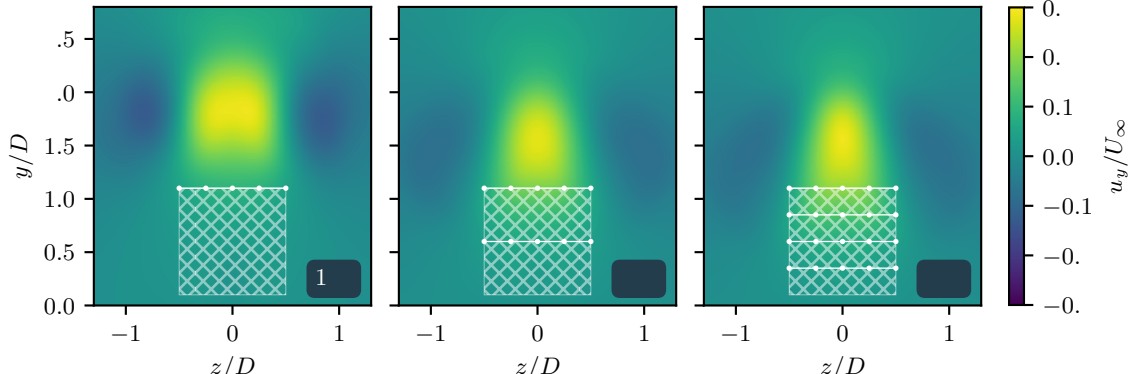

**Figure 9.** Wakes of ABL-controlled setups, colored b $u_y/U_\infty$ (rotor projected area represented by the white-hashed region, ABL-controlling
devices represented by white-dotted line segments). This figure illustrates the outcomes described by Eq. 10, where $u_y$ is at its maximum at
the $z/D = 0$ plane.

Let us extend the above analysis to configurations with multiple wings, specifically Cases 2W and 4W. From the visual
inspection of the velocity and vorticity fields above (see Figs. 7 and 8), it was concluded that the advection of low-momentum
fluid parcels is more efficient in setups with a greater number of wings. This assertion is revisited in light of potential flow
theory: the circulation associated with each wing-tip vortex, $\Gamma_x$, causes the vortices generated by wings at lower heights to
push those formed by the higher wings outward. Simultaneously, vortices at higher $y/D$ locations push the vortices beneath
them inward, toward the $z/D = 0$ plane. This trend is evident from the positions of the vortex cores relative to the $y$-axis on
the $x/D = 2$ panels for Cases 2W and 4W in Fig. 8. During the vortex-coalescence step, the induced velocity fields resulting
from the circulation of the top-most vortices facilitate the upward advection of the bottom-most vortices. This process is driven
by the underlying pressure field, promoting vortex coalescence. As the coalescence leads to the formation of two counter-
rotating vortices, their induction fields further enhance their mutual upward advection. This coalescence and advection occur
around $x/D = 4$ in Fig. 8. Consequently, only around $x/D \approx 4$ do the low-momentum fluid parcels, swirling within the
vortices, encounter favorable conditions to ascend to higher regions of the ABL. The requirement for the formation of these




large vortices before the entire wake can be advected upward explains why systems with more wings are more efficient at transporting low-momentum flow upwards, while simultaneously being slower at elevating the wing-tip vortices.

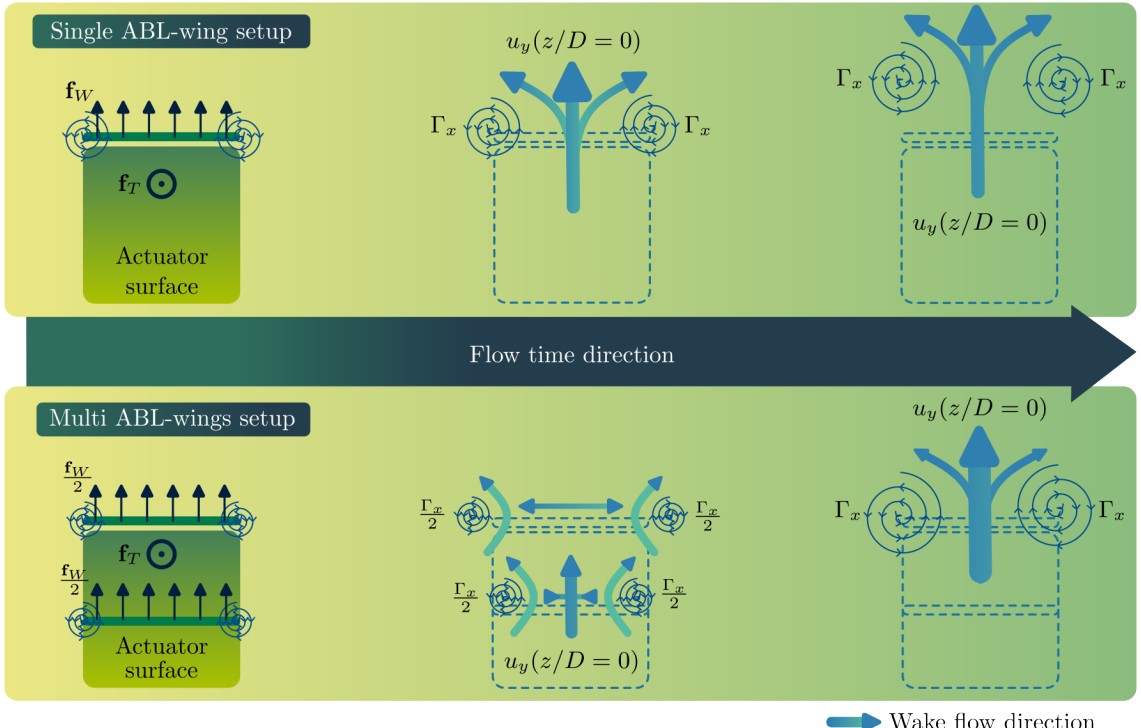

**Figure 10.** Schematic diagram illustrating the mechanisms promoting the upward advection in the wake. Setups with a single wing behind the multirotor system (represented as green line segments) move the wing-tip vortexes farther up the atmosphere. In contrast, setups with more wings advect the wing-tip vortices upwards more slowly but are more efficient in moving more low-momentum flows upwards.

Figure 10 includes a schematic diagram of the processes taking place in each ABL-controlling strategy discussed above. The momentum transfer mechanisms are divided here into twofold: setups with a single ABL-controlling wing and configurations

with multiple wings. This subdivision is based on the current results, which suggest that the mechanics of the ABL-controlling systems can generally be classified into these two categories. In the diagram, arrow thickness indicates the intensity (i.e., magnitude) of the underlying velocity field, while vortex cores are depicted as spirals. The flow evolution is represented in three stages for both cases.

### 3.3    Momentum entrainment

In the previous subsection, the topology of the wakes of the ABL-controlled systems was analysed using the velocity and vorticity fields. This subsection focuses on the momentum and energy balances of the ABL-controlled flows for the different design strategies of Cases 1W, 2W, 4W, 1W-ND, 2W-ND, 4W-ND.





Figure 11 presents the velocity products at a horizontal plane at $y/D = 1.1$ to qualitatively assess the ABL-controlling effects on vertical momentum flux and the enhancement of power extraction. Negative values of the product $-u_y u_x$ indicate momentum transfer from below to above the plane, while positive values indicate transfer from above to below the plane. The results shown in Fig. 11 align with previous observations from the velocity fields in § 3.2, showing that all ABL-controlled setups significantly increase vertical momentum exchange between the ABL and the wake. This increase is expected, given the higher vertical velocity component observed in ABL-controlled setups.

The negative $-u_y u_x$ values in the ABL-controlled configurations suggest that low-momentum flow parcels in the wake are advected upward. Conversely, positive values indicate that high-momentum flow from the ABL is entrained into the wind-farm flow from the sides of the wake ($|z/D| > 0.5$). The $-u_y u_x$ fields also demonstrate a significant performance gain when using the 2- or 4-wing design compared to the single-wing design of Case 1W. This improvement arises from a more uniform distribution of circulation $\Gamma_x$ across the wake. This uniformity reduces the average distance between the rotation center of the wing-tip vortex and the low-momentum flow parcels in the wake, thereby enhancing momentum fluxes across the $y/D = 1.0$ plane.

Figure 12 displays the velocity products $u_z u_x$ from a side-plane view at $z/D = 0.5$. Positive values indicate momentum transfer towards the $-z$ direction (out of the page plane), while negative values indicate the opposite. In setups with a single wing, such as Cases 1W and 1W-ND, momentum exchange is predominant in the near wake region and is primarily influenced by the tip vortices of the single ABL device. In configurations with more wings, the momentum exchange between the wake and surrounding flows is distributed among the multiple ABL-controlling devices. The $u_z u_x$ plots also reveal that regions with positive momentum entrainment (brightly-coloured regions) are aligned behind the disk actuator in setups with more wings. This suggests that while configurations with more wings may not be as effective in quickly advecting wing-tip vortices upwards, they are more efficient at pushing high-momentum flow into the wind-farm flow from the sides.

The comparison of momentum fluxes between the wake and the surrounding flows for models with and without the induced drag from ABL-controlling wings reveals that the impact of induced drag depends on how it is distributed across the ABL-controlling system. For instance, the results in Fig. 11 and 12 demonstrate that higher momentum fluxes occur when induced drag is concentrated on a single wing (as seen in Cases 1W and 1W-ND). In setups with multiple wings, where the induced drag $C_{x,W}$ is distributed over a larger wake region, the induced drag has a less significant effect on the dynamics of the ABL-controlled flows.

When the induced drag is concentrated at the top wing element (at $y/D = 1.1$)there are larger velocity gradients in the near-wake region of that wing. These gradients, combined with the shearing processes in the outer layers of the wake flow, lead to more intense vorticity production and consequently, more vigorous turbulent mixing. In configurations with more wings, this effect is spread over a larger region of the wake, making the impact less pronounced.

Lastly, the efficiency of the current ABL-controlling strategy is also assessed through the total pressure, $p_t \approx p_t(x)$,





**Figure 11.** Products of vertical and streamwise velocities at the top of the disk actuator model (represented as a white box) plane, $y/D = 1.1$.

$$p_t(x) := p(x) + \frac{1}{2}\rho \sum_{i=1}^{3} \bar{\mathbf{u}}_i(x)^2 \tag{11}$$





**Figure 12.** Products of crosswind and streamwise velocities at a plane at the left hand side of the disk actuator model (represented as a white box) plane, $z/D = 0.5$.

available in the wake. The total pressure is integrated inside a flow volume defined by the projected cross-sectional area of the actuator disk surface, i.e., $\{x \in [0, +\infty[; y \in [0.1D, 1.1D]; z \in [-0.5D, 0.5D]\}$. The results for the total pressure along the wake, $p_t(x)$, are shown in Fig. 13.





According to the Momentum theory, the total pressure behind an actuator disk is given by:

$$p_t(x=0) = p_\infty + \frac{1}{2}\rho((1-2a)U_\infty)^2 \tag{12}$$

where $a$ is the induction factor, $a := (U_\infty - u_x(x=0))/U_\infty$. According to the Momentum theory, the induction factor $a$ relates to the actuator disk's thrust coefficient $C_T$ by the equation $C_T = 4(1-a)$. For the present system, Momentum theory predicts the total pressure in the wake to be approximately $p_t/p_{t,\infty}(x/D > 0) \approx 0.28$. However, the current viscous wake model can recover momentum only through velocity fluctuations around the streamtube (as seen in the Baseline case). When the ABL-

345 controlling strategy is applied, wake recovery is significantly faster due to wake-steering, and turbulent velocity fluctuations become a secondary mechanism for momentum entrainment and wake recovery. Consistent with the results shown in Figs. 11 and 12, the total pressure integrated along the wake indicates that design strategies with more wings expedite wake recovery. For instance, in Cases 2W or 4W, the total pressure in the wake reaches 95% of the free-stream value at $x/D \approx 5$ and 6, respectively. In contrast, for the Baseline case, which lacks ABL-controlling devices, wake recovery is substantially slower,

and this level of recovery is not observed within the current computational domain extending to $x/D = 50$.

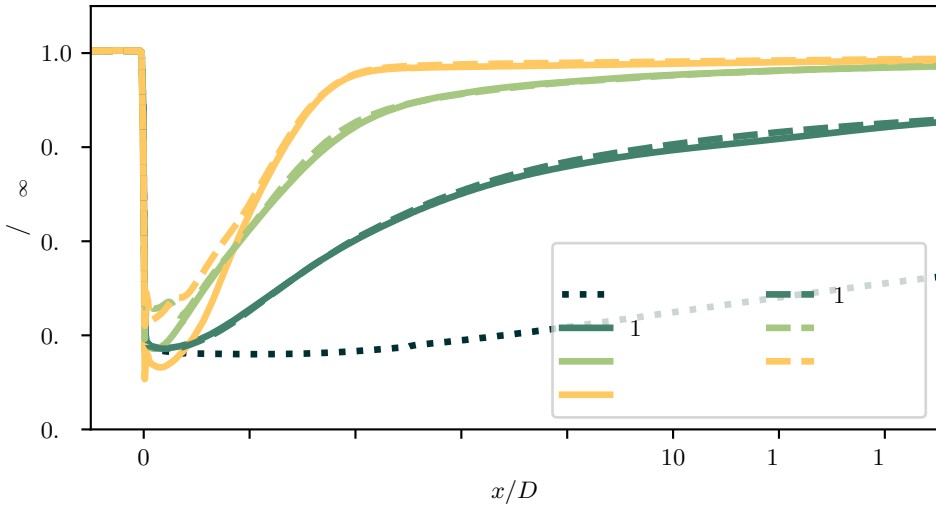

**Figure 13.** The figure illustrates the total pressure available in the wake as a function of the streamwise flow direction, $x/D$ for all test cases in this study. The total pressure is integrated within a box-shaped volume corresponding to the rotor's projected area in the downstream direction. Dashed and solid curves represent ABL-controlled setups with and without the wing's induced drag, respectively. The dotted line illustrates the results for the system without ABL-controlling features.

The analysis of the total pressure integrated along the wake is repeated for the cubed velocity, $\sum_i u_i^3(x)$. The results of this analysis are shown in Fig. 14. While total pressure quantifies the energy available in the wake, the cubed velocity is a more appropriate parameter for measuring the wind power available for extraction in the wind farm flow. The cubed velocity plots in Fig. 14 reveal a significant improvement in power recovery when the ABL-controlling system is used. Comparing the Baseline



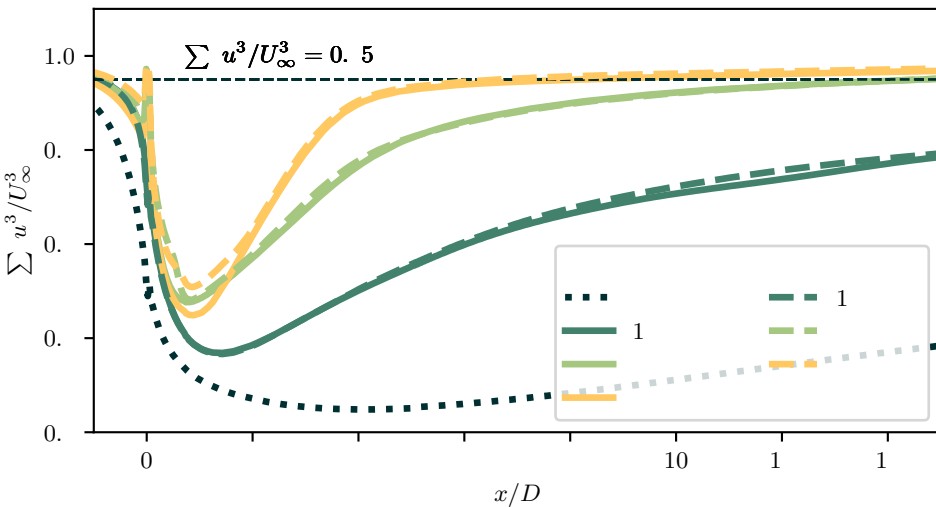

**Figure 14.** The figure displays the cubed velocity for all test cases in this study. The dashed lines represent systems without induced drag from the wings, whereas the solid lines represent systems with induced drag from the wings. The dotted line illustrates the results for the system without ABL-controlling features. The thin dashed line indicates the 0.95 plateau for reference.

case to Cases 1W and 1W-ND shows this enhancement clearly. A similar improvement is observed when comparing the setup with two wings to the single-wing setup. Finally, the system with four wings, which demonstrates the fastest wake recovery among the setups analyzed, recovers to 95% of the free-stream velocity at $x/D = 6$ downstream of the multirotor system. This underscore the potential of the technology compared to current technologies without ABL-controlling devices.

## 4  Conclusions

The current study introduces and evaluates a novel concept of multirotor wind farm layouts, incorporating paired rotor and rotor-sized wings termed ABL-control devices, situated in the near wake region. These rotor-sized wings generate vortical structures within the wind farm flow, enhancing the vertical momentum flux from the flow above the ABL into the wind farm flow. This augmentation facilitates the wake-recovery process, leading to possible increased power generation per land area. Multirotor and ABL-controlling devices are characterized using three-dimensional actuator surface models based on

Momentum theory. Analysis of velocity and vorticity fields reveals that large wing-tip vortices are responsible for advecting low-momentum fluid parcels from the wake flow upwards in exchange for moving high-momentum flow from the ABL downwards. Additionally, it was observed that the induced drag of the large wings comprising the ABL-controlling systems could slightly enhance the mixing process at the outer shear layers of the wake for setups with significant and concentrated induced drag forces. Hence, the current findings suggest that the induced drag of the wings may favor the ABL-controlling strategy.

Furthermore, the examination of momentum flux and total pressure indicates that, with the adoption of ABL-controlling strategies, the vertical momentum flux becomes the primary mechanism for wake recovery, while velocity fluctuations assume a





secondary role for the assessed flow conditions. For the four-winged ABL-controlling strategy, the total pressure and power in the wake recovered to 95% of the free-stream value at downstream positions of approximately $x/D \approx 5$ and $x/D \approx 6$, respectively. These results underscore the technology's potential to reduce the land area required for wind farms.

*Code availability.* The primary codebase which enables the multi-rotor system simulations with vortex-generating modes, is hosted on GitHub. You can access it via the following link: https://doi.org/10.5281/zenodo.11615669. This repository includes detailed documentation and example scripts to facilitate replication and extension of our work.

*Data availability.* The database presented in this study is available from the corresponding author upon reasonable request. The database is compressed in Python's Pickle library and relies on several open-source libraries, including NumPy, Pickle, SciPy, and Matplotlib.

*Author contributions.* FM: formal analysis, writing, code development. AZ: supervision and technical review. CF: funding acquisition, supervision and technical review.

*Competing interests.* The contact author has declared that none of the authors has any competing interests.

*Acknowledgements.* The authors would like to express their sincere gratitude to Thomas Broertjes for providing them with the Computer-Aided Design (CAD) model of the multirotor system.EXTT



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
