# Peer review of "Proof of Concept for Multirotor Systems with Vortex-Generating Modes for Regenerative Wind Energy: A Study Based on Numerical Simulations and Experimental Data"

_Wind Energy Science, 2024_

## Referee Comment (RC1)

General Comments:

Overall, the work is novel and exciting, relating simulations to potential flow theory to good effect. I foresee this being a high-quality publication, however several shortcomings need major attention:

- In the Introduction, there is confusing terminology as "wing" is often used in place of "blade" when referring to the actual blades of the turbine. "Wing" should be reserved for the stationary lifting element mounted behind the rotating blades.
- The review of previous literature is erratic and compromised by confusing use of the term "wing-pitch" and "blade-pitch" in inappropriate places (i.e., to describe wake steering, for instance). I recommend adopting a more conventional naming scheme (static induction control, wake steering, wake mixing, for instance) or justifying the use of your new language.
- Throughout the manuscript (i.e., all figures after Fig 3) axis labels and colorbar labels are missing values, making interpretation of the figures impossible.
- The grid convergence study was done on a configuration without ABL control devices. However, a main aspect of the article relates to the use of ABL control devices, which produce significant vortices in the wake. No convergence study is done on the ABL-controlled configuration nor is even given a description of how the mesh size relates to the size of the streamwise vortices generated (i.e., how many cells are expected across the smallest vortex to be generated? And are there any limitations to acknowledge because of this value?).
- At first read, it's not clear why so much attention is paid to comparing the ND and non-ND cases. The first mention of "induced-drag" is in Section 3.2 when describing the layout of results figures. The significance of including vs not including ND in the simulation must be introduced first.

Specific Comments:

Line 22 – why limit this to "per land surface area"? Land is mentioned twice in this paragraph but it could also be sea surface area.

Line 31 – why are you excluding "blade-pitch" control strategies like the pulse and helix methods?

Line 32 – this paragraph appears haphazardly written, jumping from the reference of Dilip and Porte-Agel (isn't this a paper on wake steering not "wing-pitch control") of HAWTs to that of Nash (again, this is not "blade-pitch control" but standard wake steering that is mostly discussed) of HAWTs, right into Ferreira's work on VAWTs. This progression demonstrates lack of knowledge of the existing literature and should be cleaned up and expanded.

Line 36 – What do you mean "non-pitched wings"? Was Ferreira using pitched "wings" (i.e., blades)? Maybe it is implied in the next sentence, but the reader shouldn't have to infer what you mean.

Line 45 – this paragraph again is haphazard. There have already been several references to yaw-control strategies (i.e., wake steering) of HAWTs and VAWTs, and now the text is re-describing wake steering in the HAWT context.

Line 111 – is the use of D^2 implying that the multi-rotor system has the same width as height? If so, could you mention that explicitly?

Line 122 – it appears that the multirotor system is spread over many cells in Fig. 3; why does the text say that the multirotor extends "one finite-volume cell" in the vertical direction?

Line 147 – there is no circular geometry in this setup, so how does a "diameter-based Reynolds number" make sense here?

Line 149 – please comment on the realism of the 1% freestream turbulence intensity and how this value may affect conclusions to be drawn

Line 157 – if $f_T$ and $f_W$ are kept constant and $C_{y,W}$ and $C_{x,W}$ are also always constant, then is the wing's chord being reduced as the number of wings increase? Some mention of how this is accomplished is merited.

Line 231 – why, after splitting the ND cases from the non-ND cases in Figures 6-7, does Figure 8 combine them into one figure (without any mention in the figure caption of where the ND and non-ND cases are in the figure)? Line 253 would be a good place to introduce a new Figure 9.

Line 278 – again, there is no indicator in the figure of which three cases we are looking at

Line 300 – it would be helpful to introduce the $u_y u_w$ quantity here before discussing the figures to give context why this quantity is relevant and why $u'_y u'_w$, which is often the entrainment quantity of interest behind a wind turbine, is not considered. Brief discussion about how the baseline case will still get most of its wake recovery from the turbulent flux might be helpful, too.

Technical Corrections:

Line 1 – please change "pared" to "paired"

Line 89 – deferrable is misspelled

Line 124 – remove the word "far"

Line 145 – the sentence "The outlet is modeled as a free-stream flow condition and zero-gradient for pressure" has already been used earlier.

Line 149 – please change the word "verging"

Line 159 – please define "ND" before using in the table

Line 164 – cell elements do not have a "diameter"

Line 240 – "threefold" (what?), and again in line 294 "twofold" is used without a noun following it

Line 267 – would it be more readable to say $u_y(z)$ instead of $u_y(z/D)$?

---

## Referee Comment (RC2)

**A Numerical Investigation of Multirotor Systems with Vortex-Generating Modes for Regenerative Wind Energy: Validation Against Experimental Data by Martins et al.**

August 1, 2024

The authors aim to provide a proof of concept of increasing the wake recovery of a multi-rotor system with vertical axis wind turbines using flow control devices, which is an interesting topic for the wind energy community. The authors perform Reynolds-averaged Navier-Stokes (RANS) simulations of a uniformly distributed thrust load in the form of a square, with and without additional vertical body forces, subjected to a uniform, laminar inflow. The simplified setup is validated qualitatively with wind tunnel PIV measurements. Based on the RANS simulations, the authors conclude that the wake recovery of the multi-rotor system can be enhanced by an order of magnitude. However, the authors do not provide sufficient evidence to conclude that the wake recovery of a utility scale multi-rotor system operating in atmospheric turbulence can be enhanced due to the chosen laminar inflow. Furthermore, the authors do not provide information of the power coefficient of the multi-rotor system with VAWTs nor wind farm simulation results, which may affect the author's conclusion of the potential increase in wind energy extraction per area of land. In addition, there are many labels and numbers missing in the plots, which makes it impossible for me to review the results. Therefore, I recommend a major revision. I have listed more detailed comments below.

**Main comments**

1. The authors have employed a uniform, laminar inflow for simplicity. While this is a logical first step, the obtained downstream wake recovery of the multi-rotor system is expected to be unrealistically slow and the effect of the additional atmospheric boundary layer (ABL) control devices may be exaggerated as a consequence. Figure 14 shows that the baseline case velocity deficit is about $1 - 0.2^{1/3} \approx 40\%$ at $12.5D$ downstream. (Note that I had to measure the numbers by a ruler due missing tick labels so I might have made a mistake.) This is extreme considering the employed thrust coefficient of 0.72, which translates to a maximum deficit of $1 - \sqrt{1 - 0.72} = 47\%$ following 1D momentum theory. In other words, the wake has only recovered by 7% at $12.5D$. Hence, the present results cannot be used to make conclusions about faster wake recovery for utility size multi-rotor systems operating in turbulence atmospheric conditions. Therefore, I would highly recommended to add additional numerical results including an atmospheric inflow, e.g. a neutral atmospheric surface layer with a homogeneous roughness length. I expect that such a setup will lead to a much smaller gain in wake recovery due to the effect of atmospheric turbulence, possibly by an order of magnitude. If additional simulations are not possible then the authors should modify the abstract and conclusion such that the reader cannot be tempted to translate the present results directly to the real world application. For example, the authors could add: *The present article employs a simple numerical setup to provide a proof of concept and the current results cannot yet be translated to real world applications. Additional research is required to study the wake recovery benefit in atmospheric inflow conditions.* The authors could also consider to choose a more clear title, for example: *A Proof of Concept of Multirotor Systems with Vortex-Generating Modes for Regenerative Wind Energy based on Numerical Simulation and*

*Experimental Data* Finally, the term *ABL control device* is misleading since the simulated inflow and wind tunnel experiment do not represent an ABL.

2. The authors conclude that the land usage could potentially be reduced due to the enhanced wake recovery. While this a logical extrapolation of the present simulation results, the actual benefit of the enhanced wake recovery in wind farms may be much smaller, following the reviewer's experience with wake recovery benefits of horizontal axis turbine multi-rotors in isolation and wind farms [van der Laan et al(2019), van der Laan and Abkar(2019)]. This is because the wake added turbulence typically reduces the benefit of wake recovery devices for downstream turbines, and there are many inflow cases where a wind farm does not experience significant wake effects (non-aligned inflow wind directions, above rated inflow wind speeds.) If the authors desire to make conclusions about reduced land usage, then additional wind farm simulations of multi-rotor systems are required, including a more realistic atmospheric inflow as discussed previously. Furthermore, information on the power coefficient of the multi-rotor VAWT systems are required in order to compare with the spacing of traditional wind farms containing horizontal axis wind turbines in terms of net power density.

3. Equation 4, how is $I_{u,\infty}$ defined? If it is based on the TKE, as Eq. (4) suggested, I would advice the authors the write $I_{k,\infty}$, as $I_{u,\infty}$ could be mistaken for the streamwise component, i.e. $I_{u,\infty} = \sigma_u/U$.

4. Section 2.1: The vertical axis turbines (VAWTs) are represented by a uniformly distributed source term in the form of a square. This choice could affect the conclusion of the article since the interaction of the VAWTs and VAWT generated turbulence are expected to influence the wake recovery. The authors have compared the simplified RANS setup with wind tunnel PIV measurements in Figs. 4 and 5, which provides a qualitative validation. The authors do not report a quantitative comparison and this makes it difficult to understand the errors in the simplified RANS model. I would strongly recommend the authors to add this. In addition, have the authors verified the simple uniformly distributed source term model with a numerical model where closely spaced VAWTs are represented by a high fidelity model, as for example an actuator line model? Such a comparison would provide useful information on the errors introduced by the simplified model.

5. There are many labels and numbers missing in the plots, which makes it impossible for me to review the results of Section 3 in detail.

**Minor comments**

1. Introduction, Line 26: *With the characteristic height of the ABL around one kilometer, wind farms covering a surface area of over 1020 km can approach the asymptotic limit of "infinite" wind farms.* Do the authors miss a "-" sign as in 10-20 km?

2. The authors depict the measured VAWT system by a diagonal white lines in Figs 4 and 5. This is confusing and the reader could think that the measurements actually represent a turbulence grid instead of VAWTs. The authors could add an explanation in the figure captions.

**References**

[van der Laan and Abkar(2019)] M. P. van der Laan and M. Abkar. Improved energy production of multi-rotor wind farms. *J. Phys.: Conf. Ser.*, 1256(012011):1–11, 2019. doi: 10.1088/1742-6596/1256/1/012011.

[van der Laan et al(2019)] M. P. van der Laan et al. Power curve and wake analyses of the vestas multi-rotor demonstrator. *Wind Energy Science*, 4:251–271, 2019. doi: 10.5194/wes-4-251-2019.

---

## Author Comment (AC1)

**Review feedback**

Below, the authors summarize the points raised by the reviwers and address them individually.

**First reviewer comments:**

1. Overall, the work is novel and exciting, relating simulations to potential flow theory to good effect. I foresee this being a high-quality publication, however several shortcomings need major attention
   - Reply: The authors thank the reviewer for recognizing their work. They have carefully considered the reviewer's comments and have made every effort to address them. Below, the reviewer will find the authors' responses to each of the suggestions provided.
2. In the Introduction, there is confusing terminology as "wing" is often used in place of "blade" when referring to the actual blades of the turbine. "Wing" should be reserved for the stationary lifting element mounted behind the rotating blades.
   - Reply: The authors shortened the introduction chapter prior to submission, which led to some confusion in terminology. We have since revised the introduction to ensure that the terms "blade" and "wing" are clearly distinguished.
3. The review of previous literature is erratic and compromised by confusing use of the term "wing-pitch" and "blade-pitch" in inappropriate places (i.e., to describe wake steering, for instance). I recommend adopting a more conventional naming scheme (static induction control, wake steering, wake mixing, for instance) or justifying the use of your new language.
   - Reply: The authors thank the reviewer for the advice and acknowledge noticing the same issue. They have addressed this by adopting a more consistent naming convention.
4. Throughout the manuscript (i.e., all figures after Fig 3) axis labels and colorbar labels are missing values, making interpretation of the figures impossible.
   - Reply: The issue with the figures was likely due to a technical problem encountered while compiling our vector files online. To prevent this from recurring, the authors have replaced all figures with high-resolution raster files. Additionally, the colors of the figures have been adjusted to improve readability in printed formats.
5. The grid convergence study was done on a configuration without ABL control devices. However, a main aspect of the article relates to the use of ABL control devices, which produce significant vortices in the wake. No convergence study is done on the ABL-controlled configuration nor is even given a description of how the mesh size relates to the size of the streamwise vortices generated (i.e., how many cells are expected across the smallest vortex to be generated? And are there any limitations to acknowledge because of this value?).

- Reply: The authors have further elaborated on the grid convergence (i.e., grid sensitivity) analysis in the revised version of the manuscript. The reviewer can find this discussion starting on line 189. In this section, the authors comment on the grid resolution with respect to solving for the blade-tip vortices. Additionally, Table 2 shows that the adopted grid resolution (Fine) is sufficient for obtaining grid-insensitive results, which is expected since the model only addresses the largest scales in the flow.

6. At first read, it's not clear why so much attention is paid to comparing the ND and non-ND cases. The first mention of "induced-drag" is in Section 3.2 when describing the layout of results figures. The significance of including vs not including ND in the simulation must be introduced first.
   - Reply: The authors agree with the reviewer and have addressed this suggestion by including comments on the necessity of isolating the effect of the wing's induced drag from the resulting induction field behind the multirotor system. These comments can be found on lines 95 and 176 of the revised manuscript.

7. Punctual feedbacks:
   - Line 22 – why limit this to "per land surface area"? Land is mentioned twice in this paragraph but it could also be sea surface area.
     - Reply: The authors have clarified this point by adding "land or sea surface area."
   - Line 31 – why are you excluding "blade-pitch" control strategies like the pulse and helix methods?
     - Reply: The authors have included a brief discussion on the two methodologies.
   - Line 32 – this paragraph appears haphazardly written, jumping from the reference of Dilip and Porte-Agel (isn't this a paper on wake steering not "wing-pitch control") of HAWTs to that of Nash (again, this is not "blade-pitch control" but standard wake steering that is mostly discussed) of HAWTs, right into Ferreira's work on VAWTs. This progression demonstrates lack of knowledge of the existing literature and should be cleaned up and expanded.
     - Reply: The authors thank the reviewer for pointing out the inconsistencies in the introduction. Prior to submission, some text was removed, which may have compromised the quality of the introduction section. The authors have addressed this issue by rewriting the introduction to improve clarity.
   - Line 36 – What do you mean "non-pitched wings"? Was Ferreira using pitched "wings" (i.e., blades)? Maybe it is implied in the next sentence, but the reader shouldn't have to infer what you mean.
     - Reply: Please, see the answer above.
   - Line 45 – this paragraph again is haphazard. There have already been several references to yaw-control strategies (i.e., wake steering) of HAWTs and VAWTs, and now the text is re-describing wake steering

in the HAWT context.

  – Reply: Please, see the answer above.

- Line 111 – is the use of D^2 implying that the multi-rotor system has the same width as height? If so, could you mention that explicitly?

  – Reply: The authors have clarified this point on line 136.

- Line 122 – it appears that the multirotor system is spread over many cells in Fig. 3; why does the text say that the multirotor extends "one finite-volume cell" in the vertical direction?

  – Reply: The text was unclear. The authors have clarified the dimensions of the system in the paragraph starting on line 143.

- Line 147 – there is no circular geometry in this setup, so how does a "diameter-based Reynolds number" make sense here?

  – Reply: The authors replaced the term by "side-length-based"

- Line 149 – please comment on the realism of the 1% freestream turbulence intensity and how this value may affect conclusions to be drawn

  – Reply: The authors have addressed this point in the paragraph starting on line 168.

- Line 157 – if $f_T$ and $f_W$ are kept constant and $C_{y,W}$ and $C_{x,W}$ are also always constant, then is the wing's chord being reduced as the number of wings increase? Some mention of how this is accomplished is merited.

  – Reply: The authors acknowledge that the sentence was confusing. They intended to convey that the effective $C_{i,W}$ of each wing changes. This point has been clarified in the paragraph starting on line 181.

- Line 231 – why, after splitting the ND cases from the non-ND cases in Figures 6-7, does Figure 8 combine them into one figure (without any mention in the figure caption of where the ND and non-ND cases are in the figure)? Line 253 would be a good place to introduce a new Figure 9.

  – Reply: The authors understand how this division may have been unclear. In Fig. 8, the authors analyze vortical structures, so it makes more sense to visualize all cases with ABL-controlling wings together, while excluding the baseline case (without ABL-controlling features). The authors have addressed this inconsistency by explaining in the text why this particular figure includes a different selection of datasets (see the paragraph starting on line 258).

- Line 278 – again, there is no indicator in the figure of which three cases we are looking at

  – Reply: The authors have clarified the text.

- Line 300 – it would be helpful to introduce the $u_y u_w$ quantity here before discussing the figures to give context why this quantity is relevant and why $u'_y u'_w$, which is often the entrainment quantity of interest behind a wind turbine, is not considered. Brief discussion

about how the baseline case will still get most of its wake recovery from the turbulent flux might be helpful, too.

- Reply: The authors have added the paragraph starting on line 325, where they introduce the velocity products and explain why they are used to evaluate momentum exchange between the wake and the surrounding flow.

- Line 1 – please change "pared" to "paired"
  - Reply: The authors have accepeted the suggestion.
- Line 89 – deferrable is misspelled
  - Reply: The authors have accepeted the suggestion.
- Line 124 – remove the word "far"
  - Reply: The authors have accepeted the suggestion.
- Line 145 – the sentence "The outlet is modeled as a free-stream flow condition and zero-gradient for pressure" has already been used earlier.
  - Reply: The authors have accepeted the suggestion.
- Line 149 – please change the word "verging"
  - Reply: The authors have accepeted the suggestion.
- Line 159 – please define "ND" before using in the table
  - Reply: The authors have accepeted the suggestion.
- Line 164 – cell elements do not have a "diameter"
  - Reply: The authors have replaced the term by "size".
- Line 240 – "threefold" (what?), and again in line 294 "twofold" is used without a noun following it
  - Reply: The authors have rephrased the paragraph.
- Line 267 – would it be more readable to say u_y(z) instead of u_y(z/D)?
  - Reply: The authors have adopted the suggested nomenclature where possible.

**Second reviewer comments:**

8. The authors aim to provide a proof of concept of increasing the wake recovery of a multi-rotor system with vertical axis wind turbines using flow control devices, which is an interesting topic for the wind energy community. The authors perform Reynolds-averaged Navier-Stokes (RANS) simulations of a uniformly distributed thrust load in the form of a square, with and without additional vertical body forces, subjected to a uniform, laminar inflow. The simplified setup is validated qualitatively with wind tunnel PIV measurements. Based on the RANS simulations, the authors conclude that the wake recovery of the multi-rotor system can be enhanced by an order of magnitude. However, the authors do not provide sufficient evidence to conclude that the wake recovery of a utility scale multi-rotor system operating in atmospheric turbulence can be enhanced due to the chosen laminar inflow. Furthermore, the authors do not provide information of the power coefficient of the multi-rotor system with VAWTs nor wind

farm simulation results, which may affect the author's conclusion of the potential increase in wind energy extraction per area of land. In addition, there are many labels and numbers missing in the plots, which makes it impossible for me to review the results. Therefore, I recommend a major revision. I have listed more detailed comments below.

- Reply: The authors thank the reviewer for taking the time to read their work and for the kind suggestions. The authors have accepted the reviewer's suggestions and have implemented them. Below, the reviewer will find the authors' direct responses to each of the reviewer's points.

9. The authors have employed a uniform, laminar inflow for simplicity. While this is a logical first step, the obtained downstream wake recovery of the multi-rotor system is expected to be unrealistically slow and the effect of the additional atmospheric boundary layer (ABL) control devices may be exaggerated as a consequence. Figure 14 shows that the baseline case velocity deficit is about $1 - 0.2^{1/3} \approx 40\%$ at $12.5D$ downstream. (Note that I had to measure the numbers by a ruler due missing tick labels so I might have made a mistake.) This is extreme considering the employed thrust coefficient of 0.72, which translates to a maximum deficit of $1 - \sqrt{1 - 0.72} = 47\%$ following 1D momentum theory. In other words, the wake has only recovered by $7\%$ at $12.5D$. Hence, the present results cannot be used to make conclusions about faster wake recovery for utility size multi-rotor systems operating in turbulence atmospheric conditions. Therefore, I would highly recommended to add additional numerical results including an atmospheric inflow, e.g. a neutral atmospheric surface layer with a homogeneous roughness length. I expect that such a setup will lead to a much smaller gain in wake recovery due to the effect of atmospheric turbulence, possibly by an order of magnitude. If additional simulations are not possible then the authors should modify the abstract and conclusion such that the reader cannot be tempted to translate the present results directly to the real world application. For example, the authors could add: The present article employs a simple numerical setup to provide a proof of concept and the current results cannot yet be translated to real world applications. Additional research is required to study the wake recovery benefit in atmospheric inflow conditions. The authors could also consider to choose a more clear title, for example: A Proof of Concept of Multirotor Systems with Vortex-Generating Modes for Regenerative Wind Energy based on Numerical Simulation and Experimental Data. Finally, the term ABL control device is misleading since the simulated inflow and wind tunnel experiment do not represent an ABL.

- Reply: The authors thank the reviewer for the kind suggestions and the additional information provided. The authors agree that investigating ABL effects is a natural next step in the current research; however, this would require a lengthy additional study beyond the scope of this initial proof-of-concept. In response, the authors have updated the abstract, the end of the introduction section, and the

closing paragraph of the results section to acknowledge the limitations of the current methodology, following the reviewer's suggestions. The authors have also adopted the reviewer's suggestion and included a modified version of the suggested title in the paper.

10. The authors conclude that the land usage could potentially be reduced due to the enhanced wake recovery. While this a logical extrapolation of the present simulation results, the actual benefit of the enhanced wake recovery in wind farms may be much smaller, following the reviewer's experience with wake recovery benefits of horizontal axis turbine multi-rotors in isolation and wind farms [van der Laan et al(2019), van der Laan and Abkar(2019)]. This is because the wake added turbulence typically reduces the benefit of wake recovery devices for downstream turbines, and there are many inflow cases where a wind farm does not experience significant wake effects (non-aligned inflow wind directions, above rated inflow wind speeds.) If the authors desire to make conclusions about reduced land usage, then additional wind farm simulations of multi-rotor systems are required, including a more realistic atmospheric inflow as discussed previously. Furthermore, information on the power coefficient of the multi-rotor VAWT systems are required in order to compare with the spacing of traditional wind farms containing horizontal axis wind turbines in terms of net power density.

   - Reply: The authors thank the reviewer for sharing their experience and perspective on this subject. They also agree that a more realistic atmospheric flow model would likely reveal a decrease in some of the numbers presented in the idealized flow model. Nonetheless, the authors would like to emphasize that this work serves as a proof-of-concept, focusing on exploring the potential of the system and the dynamics of the steered wake. Regarding the claim about decreased land usage, the experimental results corroborate the notion that the current wake-steering strategy has excellent potential for making wind farms more compact.

   - The authors acknowledge that analyzing the impact of ABL, higher-Reynolds-number flows, and wind farm flows would provide a better understanding of the proposed technology. They are currently working on such a comprehensive assessment (the reviewer is encouraged to read https://iopscience.iop.org/article/10.1088/1742-6596/2767/9/092107/pdf, for example).

   - Regarding the coefficient of performance ($C_p$), it is approximately $C_p \approx 0.3(1 - 0.3)^2$ (as seen in Fig. 13, where the induction of the multirotor is $a \approx 0.3$). A brief discussion on this topic is provided on line 181.

11. Equation 4, how is $I_{u,\infty}$ defined? If it is based on the TKE, as Eq. (4) suggested, I would advice the authors the write $I_{k,\infty}$ , as $I_{u,\infty}$ could be mistaken for the streamwise component, i.e. $I_{u,\infty} = \sigma_u/U$ .

   - Reply: The authors thank the reviewer for the suggestion and agree that the proposed notation would be more accurate. The notation

has been adjusted accordingly.

12. Section 2.1: The vertical axis turbines (VAWTs) are represented by a uniformly distributed source term in the form of a square. This choice could affect the conclusion of the article since the interaction of the VAWTs and VAWT generated turbulence are expected to influence the wake recovery. The authors have compared the simplified RANS setup with wind tunnel PIV measurements in Figs. 4 and 5, which provides a qualitative validation. The authors so not report a quantitative comparison and this makes it difficult to understand the errors in the simplified RANS model. I would strongly recommend the authors to add this. In addition, have the authors verified the simple uniformly distributed source term model with a numerical model where closely spaced VAWTs are represented by a high fidelity model, as for example an actuator line model? Such a comparison would provide useful information on the errors introduced by the simplified model.

    - Reply: The authors thank the reviewer for the suggestion and have incorporated a quantitative error analysis based on the vertical position of the blade-tip vortices. The errors between the measurements and CFD results are 1.7% in the near-wake region and increase to approximately 3.2% further downstream, using the vertical displacement of the vortices as a metric. The authors hope that these numbers and this metric, now included on line 213, are sufficient.
    - Regarding the reviewer's second recommendation, the continuation of this project will involve higher-fidelity analyses. The authors are concerned that such a comparison might be too lengthy and distracting from the main focus of this primary research. However, they agree with the reviewer that it is an interesting addition and plan to include it in future work.

13. There are many labels and numbers missing in the plots, which makes it impossible for me to review the results of Section 3 in detail.

    - Reply: The authors believe there was an issue with the online rendering of our vectorized figures. They have decided to replace all figures with high-resolution raster images for reliability.

14. Introduction, Line 26: With the characteristic height of the ABL around one kilometer, wind farms covering a surface area of over 1020 km can approach the asymptotic limit of "infinite" wind farms. Do the authors miss a "-" sign as in 10-20 km?

    - Reply: The authors thank the reviewer for noticing the typo and have corrected the text accordingly.

15. The authors depict the measured VAWT system by a diagonal white lines in Figs 4 and 5 . This is confusing and the reader could think that the measurements actually represent a turbulence grid instead of VAWTs. The authors could add an explanation in the figure captions.

    - Reply: The authors have clarified in the figure captions that the hatched region represents the multirotor's projected area.

**To-Do's based on reviwers feedback:**

- ⊠ Update all figures to raster format to prevent compilation errors. (Address point 4 of Reviewer 1 and point 13 of Reviewer 2)
- ⊠ Replace $I_{u,\infty}$ with $I_{k,\infty}$
- ⊠ Fix terminology: use "wing" only for the stationary elements.
- ⊠ Use either "wing steering" or "blade steering," and consider standardizing on "wake-steering."
- ⊠ Comment on the grid-independence analysis, including the number of cells per vortex.
- ⊠ Add a comment early in the text on the importance of accounting for the blade's induced drag (address point 6).
- ⊠ Apply suggestions from Reviewer 1 (including typos and other minor issues).
- ⊠ Respond to the 1st comment from Reviewer 2 (point 8).
- ⊠ Incorporate suggestions from the 2nd reviewer: "The present article employs a simple numerical setup to provide a proof of concept, and the current results cannot yet be translated to real-world applications. Additional research is required to study the wake recovery benefit in atmospheric inflow conditions. The authors might also consider a clearer title, such as: 'A Proof of Concept for Multirotor Systems with Vortex-Generating Modes for Regenerative Wind Energy Based on Numerical Simulations and Experimental Data.' Finally, the term 'ABL control device' is misleading, as the simulated inflow and wind tunnel experiment do not represent an ABL." (point 9)
- ⊠ Add more information on $C_P$ and $C_T$ for the reference model.
- ⊠ Add a brief comment on the vertical position of the vortices for the two models. Address point 12 from Reviewer 2.
- ⊠ Fix the minus sign on line 26.

**Legend:**

- ☐ To be done or skipped
- ⊠ Done

---

## Author Response (AR2)

The authors have correctly responded to most of my comments. The authors mentioned in their response that they included a discussion on the power coefficient around Line 178, but I could not find this.

- R: The authors appreciate the reviewer's thorough assessment and apologize for any confusion regarding the power coefficient discussion. The intended reference was to line 378, specifically the paragraph beginning on line 377, where we discuss the use of the cubed velocity as a metric for estimating the power available for extraction in the wake.

The authors may have forgotten to add additional text. In addition, the authors write in their response a 1D momentum equation for $C_p : C_p = a(1-a)^2$, is this correct? There seems to be a "4" missing or do I misunderstand something?

- R: The authors acknowledge the reviewer's attention to detail. The correct expression for the power coefficient $C_p$ in the idealized model is indeed $C_p = 4a(1-a)^2 = 0.56$.

In addition, the actually power coefficient might not follow 1D momentum theory of a horizontal axis wind turbine. Therefore, I recommend a minor revision to allow the authors to add a discussion on the power coefficient. I believe that the information on the power coefficient is important to be able to provide the reader information on the potential of reduced land use compared to traditional wind farms of horizontal axis turbines, made in the conclusion. If proper information on the power coefficient is not available, then I would recommend to remove the land use reduction statement from the conclusion.

- R: The authors accepted the reviewer's recommendation regarding the addition of a discussion on the power coefficient. We have incorporated a comprehensive analysis of the $C_p$ values across configurations, starting on line 385. In light of this addition, the final 1.5 pages of the manuscript are now dedicated to exploring power and power extraction within the proposed system. We trust that these revisions fully address the reviewer's comments on this topic.